# Utilization of screening services on cervical cancer and associated factors among female health workers in Addis Ababa, Ethiopia

**Achamyelew Melaku, Shiferaw Negash Abebe** �ORCID *, **Sofanit Haile**

Department of Obstetrics and Gynecology, Addis Ababa University, Addis Ababa, Ethiopia

* shiferaw.negash@aau.edu.et, shiferaw_negash@yahoo.com

## Abstract

### Background

Cervical cancer is the fourth most common cancer among women worldwide and the second most common cancer in women in Ethiopia with the disease claiming the lives of more than 340, 000 women globally in 2020. A well organized and arranged screening is one of the key intervention strategies in the reduction of the incidence and mortality from the disease. Healthcare workers, being the front line in health delivery system, are expected to play a critical role in cervical cancer screening. This being the fact on the ground, the gap on the cervical cancer screening service utilization and the factors influencing it among female health workers in Ethiopia is not well elucidated. We aim to explore the magnitude of the screening uptake and associated factors among female healthcare providers in Addis Ababa, Ethiopia.

### Methods

Institution-based cross-sectional study using stratified multi-stage sampling technique was done from June 05 to July 05, 2023 among female health workers in Addis Ababa. The data was collected using a structured self-administered questionnaire. Descriptive statistics like mean, median, and proportion were used to summarize the data. Bivariable regression analysis was used to measure the association between the dependent and independent variables, while multivariable regression analysis was used to determine the independent associations. Variables with P-value≤ 0.25 on bivariable model were entered into a multivariable logistic regression model. Odds ratios with 95% confidence intervals (CIs) were computed to measure the strength of association, and statistical significance was declared at P< 0.05.

### Results

A total of 432 study participants were enrolled in the final analysis with the response rate of 100%. Out of the total participants, 243 (56.3%) were nurses, and 183 (42.4%) were of age 30 years and above. In this study, only 19.4% (84/432) and 31.2% (57/183) among all participants and the targeted groups (age ≥30 years), respectively, have utilized

**Data availability statement:** All relevant data are within the manuscript and its Supporting Information files.

**Funding:** This manuscript was partially funded by the Addis Ababa University Office for Graduate studies. The rest of the fund was equally shared among the authors. The funder for this manuscript, Addis Ababa University Office for Graduate Studies, had no role in study design, data collection and analysis, decision to publish, or preparation of the manuscript.

**Competing interests:** The authors have declared that no competing interests exist.

**Abbreviations:** CCS, Cervical cancer screening; FHWs, Female health workers; HO, Health officers; IESO, Integrated emergency surgical officers; VIA, Visual inspection with acetic acid; HFs, Health facilities; HIV, Human immunodeficiency virus; KAP, Knowledge, attitude and practice.

the screening services. The lack of attention was the main reason identified hindering the screening service utilization (32.5%) while the promoter factors being awareness about screening methods (48.8%) and physician recommendation (26.2%). On Multivariable model; participant age ≥30 years (AOR=1.6, 95%CI1.15–3.37), being married (AOR=6.1, 95%CI 2.42–15.06), and working in cervical cancer screening units (AOR=3.7, 95%CI1.01–12.12), respectively had an independent association with the screening service utilization. Similarly, the study participants knowledge of the etiology, knowledge of cigarette smoking as risk factor, and visual inspection with acetic acid as screening method had shown an independent association with screening service utilization, (AOR=1.6, 95%CI=1.01–12.12), (AOR=4.1, 95%CI1.68–9.76), (AOR=14.2, 95%CI3.77–53.32), respectively.

## Conclusion

The low screening services utilization among the targeted age group of female health workers is alarming. The lack of attention and feeling of healthy were hindering factors among those not yet screened, while those screened were motivated by the awareness created and physician recommendation. Continual awareness creation and training of female healthcare providers on cervical cancer and its screening is recommended to improve the screening service uptake by the women in the population.

## Introduction

Cervical cancer is an important public health problem invariably caused by persistent infection with high-risk human papilloma virus. Despite being largely preventable through vaccination against high-risk human papillomavirus (HPV) and regular screening programs, cervical cancer remains a global health challenge [1–5]. The incidence and mortality varied widely and geographically and globally around 604, 127 new cases and 341,831 deaths reported in 2020. In sub-Saharan Africa, 34.8 per 100,000 and 22.5 per 100,000 new cases and deaths occurring annually from the disease compared with 6.6 and 2.5 per 100,000 new cases and deaths in North America [1,2,6,7]. The drastic differences are mainly explained by lack of access to HPV vaccination, regular screening, and the high burden of HIV infection in this region [1,3–7]. The problem is still overwhelming due to scarcity of resources, deficiency of health care system, making cervical cancer screening opportunistic rather than being organized and largely relying on visual inspection with acetic acid (VIA). The low rates of cervical cancer screening for few African countries were revealed by various studies; Uganda 7%, Nigeria 8.3%, Kenya 6%, and Ethiopia 2%, respectively [8–13]).

Ethiopia has a population of 36.9 million women with ages ≥15years and who are potentially at risk of cervical cancer. Around 9.8 million women are of ages between 30–49 years who happens to be the targeted audiences for regular screening. Annually, around 7,619 new cases and 6081 deaths were reported in Ethiopia in 2020 [8,14,15]. VIA as single-visit approach combined with access to cryotherapy was piloted and introduced in Ethiopia by the Federal Ministry of Health (FMOH) in collaboration with Pathfinder International Ethiopia. This later integrated into a comprehensive care package for people living with HIV/AIDS and availing the service in 25 healthcare facilities in the country [16]. Ethiopia has adopted the new World Health Organization (WHO) cervical cancer prevention and control guideline and has implemented an arranged screening to women of ages between 30–49 years, which can be

expanded to 25–59 years on specific indications (being HIV positive or immune compromised state, have never been screened between 30–49years, population risk, resource availability). Nationally, 33% of the public health facilities including Addis Ababa provide a free cervical cancer screening as of 2018. In Addis Ababa, the capital city of the country, all public health centers and hospitals have been providing cervical cancer screening services as of 2020 [17]. However, the coverage of cervical cancer screening in the country remains very low, ranging from 2.0–20.2% and 0.4–14.0% in the urban and rural areas, respectively [1,9,14,16,18].

Health workers are the trusted source of medical information and their attitudes and practices towards diseases of public health importance, like cervical cancer, are very critical if the country must make progress towards the prevention of such diseases [19, 20]. As shown in various studies, Female Health Workers (FHWs) have a high level of knowledge and positive attitude towards cervical cancer screening (CCS). However, CCS uptake remained very low [14,18,21,22]. Healthcare providers in hospitals and health centers constitute the most visible front-line personnel providing health education to patients and the general population. In addition, female healthcare providers play an integral role in educating and awareness creation of women in the prevention of diseases. They can also influence cervical cancer screening adherence and health promotion, and act as a role model to the women in the community [23, 24].

There were many studies done on cervical cancer screening across different parts of the country with varying study population and design with the main outcome of interest being the knowledge, attitude, and practice (KAP). The healthcare providers who work at the public health institutions in Ethiopia encounter invariably all of the clients who are coming for cervical cancer screening. We have undertaken this study to address the screening service utilization as the main outcome factor and the factors influencing the uptake among female healthcare providers with different health-related disciplines and academic level and who were working at the primary and tertiary health care levels (health centers and hospitals), in Addis Ababa, and to make the study more manageable and feasible, the study was delimited to government health institutions.

## Methods

### Ethical considerations

Ethical clearance was obtained from the Ethics Review Committee of the Department of Obstetrics and Gynecology, College of Health Sciences, Addis Ababa University, with reference number DRPC 2023/02/10. Informed written consent was obtained from all individual participants included in the study. All study participants were detailed about the study and their participation was voluntary and anonymous. Since our research used human data, our study was performed in accordance with the relevant guidelines and regulations, such as principles of Helsinki Declarations.

### Study setting

The study participants were recruited from the public health institutions in Addis Ababa (AA). The city was founded in1886 by Emperor Menelik II and administratively divided into three layers: the city council at the top, 11subcities (counties) in the middle, and 117woredas (districts) at the lowest level [9]. About 25% of the country's urban population with diverse ethnic and religious backgrounds live in Addis Ababa, the current population estimate of the city being 4,794,000 [14]. The city has 10 public hospitals and 101 public health centers. As it stands to date, public health facilities are the main providers of the organized cervical cancer screening in the country.

## Study design and period

A facility based cross-sectional study was conducted from June 05,2023 to July 05,2023. The data was collected using structured questionnaire from 432 eligible female healthcare workers practicing in the selected public health institutions during the study period.

## Study participants

We recruited eligible female healthcare providers of age 25 years or more from the selected health facilities. The participants were sexually active or ever have been, never had hysterectomy nor diagnosed with cervical cancer and comprised of doctors, nurses, midwives, pharmacists/pharmacy technicians, medical laboratory technologists, environmental health professionals, nutritionists, and public health officers. There were 9,509 female healthcare professionals practicing in the public health facilities in AA (HCs and Hospitals) during the study period; of these 5,715 were clinical nurses, 1,227 medical doctors, 1,172 midwives, 937 public health officers (HO) and integrated emergency surgical officers (IESO), and 457 constitute others. All participants in the selected healthcare facilities had equal chance of enrolment in the study. The minimum age of 25 years was considered to maximize representativeness and there are also some recommendations to start screening for the general population at age of 25 years [24].

## Sample size and sampling procedure

From 101 public health centers, 64 health centers were selected using stratified multi-stage sampling. The sample size calculation was adjusted to get the minimum allowable representation from the designated profession (department/working unit) within the selected health institution which in our case is the sampling unit (S1 Fig). The Addis Ababa city has 11 sub-cities (counties) and the health centers are divided in to two groups "A" or "B" based on the specific services served. For our case since cervical cancer screening is done in both groups, we have included all 101 health centers. The two out of the six public hospitals were selected using simple random sampling technique. A single population proportion formula was used to calculate the sample size: $n = (z^2pq/d^2)$, where n is the desired sample size; z is 1.96; p is the proportion of the national cervical cancer screening utilization which is (~15%) from systematic review and meta-analysis, the pooled national cervical cancer screening utilization was 14.79% (95% CI: 11.75, 17.83) [25]; and d is the level of precision set at 0.05. Accordingly, the calculated total sample size was 196, and by adding a 10% non-response rate, the sample size became 216. Considering the design effect of 2, the final sample size obtained was 432.

**Study Variables.** Socio-demographic variables, years of work experience, type of profession, place of work, monthly household income, and the perceived influencing factors (knowledge and attitude). The cervical cancer screening service utilization was the **dependent/outcome** variable of interest in this study (S2 Fig).

The sample size 'n' as per the proportional allocation for each selected healthcare unit/department was computed using the formula:

$n_i = N_i \times n/N$, where: $n_i$ =number of female health professionals that are needed from specific health unit/ department in the hospitals/HC.

$N_i$=total number of female healthcare professionals who are working within specific unit/department,

n =calculated sample size,

N = total number of female healthcare professionals in AA public health institutions.

### Data collection tools and data quality

After obtaining an informed written consent, a self-administered structured questionnaire was used for data collection. The data was collected by trained clinical nurses supervised by the investigators. The data collectors were not employee of the facility where from the data was collected. The questions aimed to gather information regarding respondent's knowledge on screening for cervical carcinoma, their attitude, practice towards screening for cervical lesion and influencing factors for utilization of screening service. Knowledge and attitude are assessed in this manuscript mainly to be used as one of the factors influencing the screening service use but not primarily for the KAP assessment intention. To ensure data quality, training was given for both data collectors and supervisor on the aim of the study and data collection procedures, and the questionnaire was pre-tested on 22 female healthcare workers (5% of total sample size) who were employed outside the selected health institutions. Moreover, the data collection process was checked by supervisor and investigator on a daily basis to ensure data completeness and consistency

### Statistical analysis

The data was checked manually for its completeness, coded, and entered into Epi-Info version 4.6 statistical package, then exported to SPSS version 25 for further analysis. Appropriate descriptive statistic was done on an individual variable. The knowledge and attitude scores were computed on the predetermined criteria in this study that participants who score above the mean to the knowledge and attitude questions were considered to have good knowledge (knowledgeable) and favorable attitude, respectively. Both bivariable and multivariable logistic regression analyses were performed to determine the association of the independent variables with the dependent variable. Variables with P-value ≤ 0.25 on bivariate model were entered into a multivariate logistic regression model. Odds ratios with their 95% confidence intervals (CIs) were computed to quantify the strength of association, and statistical significance was declared at *P*<0.05.

## Results

### Sociodemographic characteristics of the study participants

A total of 432 participants of age 25–59 years were included in the final analysis. Of the total study participants, 276 (63.9%) were working at the health centers and 249 (57.6%) were of age below 30 years. Three hundred seventy-two (86%) were non-physician, of which, clinical nurses and midwives together account for 301 (69.7%). Similarly, of the total participants 199 (46.1%) had work experience of five years or less. The household monthly income of most participants was between 5,000–10, 000birr for 248 (57.4%) followed by 10,000–50,000birr for 126 (29.2%), respectively (1USD=54.7442birr) (Table 1).

### Knowledge of the study participants on cervical cancer screening

Based on the predetermined criteria, only 234 (54.2%) of the total participants were considered to have good knowledge about cervical cancer screening. On further analysis of knowledge related characteristics, 387(89.6%) of the participants knew the etiology of cervical cancer. Most study participants described having multiple sexual partners, and early age at sexual experience as risk factor for cervical cancer, 366 (84.7%) and 307 (71.1%), respectively. However, cigarette smoking as risk factor was reported by less than half of the participants, 194 (44.9%). Vaccination followed by screening were the two most frequently mentioned preventive methods among the participants in this study, 365 (84.5%) and 296 (68.5%),

**Table 1.** The sociodemographic characteristics of the participants, July 2023.

| Variable | Frequency | Percentage |
|---|---|---|
| **Age of the participants** | | |
| < 30 years | 249 | 57.6 |
| ≥ 30years | 183 | 42.4 |
| **Religion**: | | |
| Orthodox | 284 | 65.7 |
| Muslim | 71 | 16.4 |
| Protestant | 73 | 16.9 |
| Catholic | 4 | 0.9 |
| **Marital status** | | |
| Single | 192 | 44.4 |
| Married | 234 | 54.2 |
| Divorced | 6 | 1.4 |
| **Parity** | | |
| Nulliparous | 235 | 54.4 |
| One | 69 | 16.0 |
| Two | 81 | 18.8 |
| Three | 37 | 8.6 |
| Four | 9 | 2.1 |
| Five | 1 | 0.2 |
| **Professional designation** | | |
| Physician | 60 | 13.9 |
| Non-physician health provider | 372 | 86.1 |
| **Specific types of profession** | | |
| GP | 36 | 8.3 |
| Specialist | 24 | 5.6 |
| Clinical nurse | 243 | 56.3 |
| Midwife | 58 | 13.4 |
| HO &IESO | 41 | 9.5 |
| Others[a] (pharmacist, Lab tech., Anesthetist…) | 30 | 7.0 |
| **Years of services** | | |
| ≤ 5 years | 199 | 46.1 |
| 6–10 years | 171 | 39.6 |
| 11–15 years | 44 | 10.2 |
| >15 years | 18 | 4.2 |
| **Working unit** | | |
| Obstetrics and gynecology (ANC, LW, in-patient, Gyn OPD) | 94 | 21.8 |
| Cervical cancer screening room | 33 | 7.6 |
| ART &TB | 24 | 5.6 |
| Other than OBGYN (OPD, ward, ICU, OR) | 260 | 60.2 |
| Others[b] (pharmacy, Nutrition clinic, lab) | 21 | 4.9 |
| **Your partner educational status-for married** | | |
| Able to read and write | 8 | 3.4 |
| Primary | 6 | 2.6 |
| Secondary | 13 | 5.6 |
| College and above | 207 | 88.5 |
| **House hold monthly income in birr (New World Bank country classification by income, 2022)** | | |
| <5,000 | 28 | 6.5 |

*(Continued)*

**Table 1.** (Continued)

| Variable | Frequency | Percentage |
|---|---|---|
| 5,000–10,000 | 248 | 57.4 |
| >10,000–50,000 | 126 | 29.2 |
| >50,000 | 30 | 6.9 |

Others [a]: Pharmacist (8), Anesthetist (8), Nutritionist (7), Laboratory technologist (7)

Others [b]: Pharmacy (8), Nutrition clinic (6), Laboratory (6)

respectively. Among the participants who had the information about screening modality for premalignant lesion; Pap smear/cytology was the most frequently reported and HPV DNA test was the least reported, 333(82.4%) and 149 (36.9%), respectively. Nearly half, 234 (54.2%), of the participants stated the correct World Health Organization (WHO) recommended age, that is 30–49 years, for the cervical cancer screening. Interestingly, 406 (94.0%) of the study participants knew that premalignant lesion and cervical cancer can be treatable (Table 2).

Among the total participants, 272 (63%) were reported to have favorable attitude towards cervical cancer screening. Nonetheless, the screening service uptake among the total study participants and the target age groups (age ≥30years) were 84/432 (19.4%) and 57/183 (31.2%), respectively.

Table 3 and 4 below describe the screening service utilization by the participants overall knowledge and professional designation. Among the participants who were enrolled in the current study, the clinical nurses and the specialist physicians have utilized the screening services more than the rest of the professionals, 22.2% and 20.8%, respectively. However, when comparison was made among the total participants who utilized the screening services, 64.3% were clinical nurses followed by 13.1% midwife nurses.

The screening service uptake of the participants having good knowledge compared to those described as not knowledgeable was 25.2% and 12.6%, respectively. When observed among the total participants who utilized the screening services, 70.2% were knowledgeable participants (Table 4).

In this study, 84/432 (19.4%, [95%CI 19.36–19.44]) of the total participants and 57/183 (31.2% [95%CI 31.16–31.64]) among the targeted age groups were screened at least once for cervical cancer. Among those who utilized the screening service; the motivating factors were awareness about the screening methods and physician recommendation, 41(48.8%), 22 (26.2%), respectively. Ten (11.9%) were screened due to the fear persuaded by having cervical cancer patient/relative. Sixty-three (75%) of the screened participants obtained the screening service only once in the past ten years, and 66 (78.6%) were screened within the past 3–5 years. Majority, 62 (73.8%) were screened by visual inspection with acetic acid (VIA), while 20 (23.8%) obtained the pap smear test.

As shown in S3 Fig, among the total participants who were not ever screened (n=348), the two most common reasons were assumption of feeling of healthy/ low risk perception 117 (33.6%) followed by lack of attention 96 (27.6%). Similarly, the lack of attention followed by the feeling of healthy were the two most common reasons for not being screened among the targeted age groups (Table 5).

## Factors affecting cervical cancer screening utilization among the study participants, July 2023

As depicted in Table 6, some variables showed significant associations with the cervical cancer service utilization on bivariable model only, while other variables showed consistent association both on bivariable and multivariable regression models. Compared with their counter parts; working at health center COR 95% CI, 1.9 (1.11–3.26), having positive attitude

**Table 2. The knowledge related characteristics of the study participants about cervical cancer and its screening (n=432), July 2023.**

| Variable | Frequency | Percentage |
|---|---|---|
| **Do you know HPV infection is the cause of cervical cancer?** | | |
| Yes | 387 | 89.6 |
| No | 45 | 10.4 |
| **How many participants know the risk factors for cervical cancer?** | | |
| Multiple sexual partners | 366 | 84.7 |
| Early sexual intercourse experience | 307 | 71.1 |
| HPV infection (Persistent) | 313 | 72.5 |
| Smoking cigarettes | 194 | 44.9 |
| **How many participants know cervical cancer prevention methods? (n = 432)** | | |
| Vaccination | 365 | 84.5 |
| Behavioral change | 228 | 52.8 |
| Screening | 296 | 68.5 |
| Early detection and treatment of pre-cancerous cervical lesions | 257 | 59.5 |
| **Have you ever heard of premalignant cervical lesion screening?** | | |
| Yes | 404 | 93.5 |
| No | 28 | 6.5 |
| **Who knew Among participants cervical lesion screening modalities(n = 404)** | | |
| Pap smear/Cytology | 333 | 82.4 |
| Visual inspection of the cervix (VIA) by applying acetic acid | 277 | 68.6 |
| Human papilloma virus DNA testing | 149 | 36.9 |
| **Participants who Know recommended interval of cervical cancer screening(3–5years)** | | |
| Yes | 262 | 60.6 |
| No | 170 | 39.4 |
| **Are you a candidate for premalignant cervical lesion screening?** | | |
| Yes | 320 | 74.1 |
| No | 102 | 23.6 |
| Didn't know | 10 | 2.3 |
| **Do you know the targeted age group(30–49yrs) for cervical cancer screening in our country (≥30 years)** | | |
| Yes | 234 | 54.2 |
| No | 198 | 45.8 |
| **Cervical cancer can usually be found at an early stage due to obvious symptoms** | | |
| Yes | 295 | 68.3 |
| No | 127 | 29.4 |
| didn't know | 10 | 2.3 |
| **Is it possible to treat premalignant lesion and cervical cancer?** | | |
| Yes | 406 | 94.0 |
| No | 20 | 4.6 |
| didn't know | 6 | 1.4 |
| **The method of treatment used for premalignant lesion and cervical cancer (n=406)** | | |
| Cryotherapy | 270 | 66.5 |
| Surgery | 272 | 67 |
| Chemotherapy | 279 | 68.7 |
| Radiotherapy | 168 | 41.4 |

Table 3. Subgroup analysis of screening service utilization among the participants by their profession, July 2023.

| Professional designation | Ever screened for premalignant cancer of the cervix | | |
|---|---|---|---|
| | Yes | No | Total |
| **Physician: General practitioner** | | | |
| Proportion of general practitioners ever screened or not | 5 (13.9%) | 31 (86.1%) | 36 (100.0%) |
| Proportion of general practitioners ever screened or not against the total participants | 5 (6.0%) | 31 (8.9%) | 36 (8.3%) |
| **Specialist** | | | |
| Proportion of specialists ever screened or not | 5 (20.8%) | 19 (79.2%) | 24 (100.0%) |
| Proportion of specialists ever screened or not against the total participants | 5 (6.0%) | 19 (5.5%) | 24 (5.6%) |
| **Nurse** | | | |
| Proportion of nurses ever screened or not | 54 (22.2%) | 189 (77.8%) | 243 (100.0%) |
| Proportion of nurses ever screened or not against the total participants | 54 (64.3%) | 189 (54.3%) | 243 (56.3%) |
| **Midwife** | | | |
| Proportion of midwives ever screened or not | 11 (19.0%) | 47 (81.0%) | 58 (100.0%) |
| Proportion of midwives ever screened or not against the total participants | 11 (13.1%) | 47 (13.5%) | 58 (13.4%) |
| **HO&IESO** | | | |
| Proportion of HO &IESO ever screened or not | 7 (17.1%) | 34 (82.9%) | 41 (100.0%) |
| Proportion of HO & IESO ever screened or not against the total participants | 7 (8.3%) | 34 (9.8%) | 41 (9.5%) |
| **Others** | | | |
| Proportion of others ever screened or not | 2 (6.7%) | 28 (93.3%) | 30 (100.0%) |
| Proportion of others ever screened or not against the total participants | 2 (2.4%) | 28 (8.0%) | 30 (6.9%) |

HO: public health officers, IESO: integrated emergency surgical officer.

Table 4. Level of knowledge and screening service utilization, July 2023.

| Knowledge Level | Ever screened for premalignant cancer of the cervix | | |
|---|---|---|---|
| | Yes | No | Total |
| **Not knowledgeable** | | | |
| Proportion of not knowledgeable ever screened or not | 25 (12.6%) | 173 (87.4%) | 198 (100.0%) |
| Proportion of not knowledgeable ever screened or not against knowledgeable | 25 (29.8%) | 173 (49.7%) | 198 (45.8%) |
| **Knowledgeable (good knowledge)** | | | |
| Proportion of knowledgeable ever screened or not | 59 (25.2%) | 175 (74.8%) | 234 (100.0%) |
| Proportion of knowledgeable ever screened or not against not knowledgeable | 70.2% | 50.3% | 54.2% |
| **Total** | 84 | 348 | 432 |
| Proportion of total ever screened or not | 19.4% | 80.6% | 100.0% |

k-score: knowledge score

COR 95% CI,1.7 (1.02–2.91), knowledge of cryotherapy as treatment method COR 95% CI, 3.6 (1.62–5.78), assertion that behavioral change COR 95% CI, 2.2 (1.32–3.62), treatment of precancerous lesion prevents cervical cancer COR 95% CI, 1.7 (1.00–2.78), knowledge of HPV DNA as screening method COR 95% CI, 1.7 (1.04–2.77), and knowledge of screening frequency COR 95% CI, 2.8 (1.62–4.99) have shown statistically significant association only on bivariable model. On multivariable logistic regression analysis; participant age ≥30 years (AOR=1.6, 95%CI1.15–3.37), being married (AOR=6.1, 95%CI 2.42–15.06), (AOR=3.7, 95%CI1.01–12.12), and working in cervical cancer screening units showed an independent association with the screening service utilization. Participants work experience ≥5 years was also shown to have statistically significant association with screening service utilization; 6–10years (AOR=3.8, 95% CI 1.52–9.12), 11–15years (AOR= 13.2, 95% CI 4.03–43.33), and

**Table 5. The reasons for those not using screening services among targeted age group (≥30 years (n = 183), July 2023.**

| Main reason not to be screened | Will you be screened if you get reminder text through your phone? | | | | |
|---|---|---|---|---|---|
| | Yes | No | Not sure | Total | Percentage |
| Lack of attention | 16 | 6 | 19 | 41 | 32.5 |
| Feeling of healthy/ the feeling of low-risk perception | 15 | 4 | 13 | 32 | 25.4 |
| Fear of pain | 12 | 10 | 6 | 28 | 22.2 |
| Lack of test awareness | 2 | 1 | 2 | 5 | 4 |
| Fear of positive result | 1 | 1 | 5 | 7 | 5.6 |
| Cost of screening | 0 | 0 | 2 | 2 | 1.6 |
| Inconvenient setup for exam | 1 | 0 | 2 | 3 | 2.4 |
| Lack of confidence in the quality of care | 2 | 0 | 6 | 8 | 6.3 |
| Total | 49 | 22 | 55 | 126 | 100.0 |

**Table 6. The bivariable and multivariable logistic regression analysis of the independent variables with cervical cancer screening utilization, July 2023.**

| Variable | Ever screen premalignant cervical cancer | | p-value | COR with 95%CI | COR with 95%CI | AOR with 95% CI |
|---|---|---|---|---|---|---|
| | Yes | No | | | | |
| **Study setting** | | | | | | |
| Health center | 63 | 213 | 0.019 | 1.9(1.11, 3.26) | 0.576 | 1.3 (0.56, 2.81) |
| Hospital | 21 | 135 | 1 | | 1 | |
| **Age of the study participants** | | | | | | |
| <30 | 27 | 222 | 1 | | 1 | |
| >30 | 57 | 126 | 0.000 | 3.7 (2.24, 6.18) | 0.05 | **1.6 (2.42, 15.06)** |
| **Marital status** | | | | | | |
| Single | 9 | 183 | 1 | | 1 | |
| Married | 74 | 160 | 0.000 | 9.4 (4.56, 19.39) | 0.000 | **6.1 (2.42, 15.06)** |
| Divorced | 1 | 5 | 0.221 | 4.1 (0.43, 38.54) | 0.668 | 2.2 (0.06. 73.46) |
| **Year of service** | | | | | | |
| <= 5 years | 13 | 186 | 1 | | 1 | |
| 6-10 years | 44 | 127 | 0.000 | 4.9 (2.57, 9.58) | 0.004 | **3.8 (1.54, 9.12)** |
| 11-15 | 21 | 23 | 0.000 | 13.1(5.78, 29.55) | 0.000 | **13.2 (4.03, 43.33)** |
| >15 years | 6 | 12 | 0.001 | 7.2 (2.31, 22.14) | 0.039 | **5.41 (1.09, 26.36)** |
| **Working unit** | | | | | | |
| Obstetrics & gynecology unit | 16 | 78 | 1 | | 1 | |
| Cervical screening unit | 21 | 12 | 0.000 | 8.5 (3.50, 20.78) | 0.035 | **3.7 (1.10, 12.78)** |
| ART & TB | 8 | 16 | 0.082 | 2.4 (0.89, 6.66) | 0.145 | 2.8 (0.69, 11.49) |
| OPD, ward & OR | 38 | 222 | 0.579 | 0.83 (0.44, 1.58) | 0.176 | 0.52 (0.21, 1.34) |
| Pharmacy, lab & anesthesia | 7 | 23 | 0.143 | 0.29 (0.23, 9.95) | 0.114 | 0.47 (0.01, 1.69) |
| **Know that HPV cause/etiology of cervical cancer** | | | | | | |
| Yes | 79 | 308 | 0.143 | 2.1 (0.78, 5.37) | 0.014 | **1.6 (1.01, 12.12)** |
| No | 5 | 40 | 1 | | 1 | |
| **Early sexual intercourse risk factor for cervical cancer** | | | | | | |
| Yes | 66 | 241 | 0.093 | 1.6 (0.92, 2.88) | 0.114 | 0.47 (0.19, 1.19) |
| No | 18 | 107 | 1 | | 1 | |
| **HPV infection risk factor for cervical cancer** | | | | | | |
| Yes | 68 | 245 | 0.054 | 1.8 (0.98, 3.23) | 0.957 | 1.1 (0.36, 2.94) |
| No | 16 | 103 | 1 | | 1 | |

*(Continued)*

**Table 6.** (Continued)

| Variable | Ever screen premalignant cervical cancer | | p-value | COR with 95%CI | COR with 95%CI | AOR with 95% CI |
|---|---|---|---|---|---|---|
| | **Yes** | **No** | | | | |
| **Smoking cigarettes risk factor for cervical cancer** | | | | | | |
| Yes | 51 | 143 | 0.001 | 2.2 (1.36, 3.61) | 0.002 | **4.1 (1.68, 9.76)** |
| No | 33 | 205 | 1 | | 1 | |
| **Behavioral change prevents cervical cancer** | | | | | | |
| Yes | 57 | 171 | 0.002 | 2.2 (1.32, 3.62) | 0.757 | 1.2 (0.46, 2.87) |
| No | 27 | 177 | 1 | | 1 | |
| **Screening is preventive method of cervical cancer** | | | | | | |
| Yes | 65 | 231 | 0.053 | 1.7 (0.99, 3.03) | 0.923 | 0.95 (0.36, 2.56) |
| No | 19 | 117 | 1 | | 1 | |
| **Early detection of precancerous cervical lesions is preventive for cervical cancer** | | | | | | |
| Yes | 58 | 199 | 0.048 | 1.7 (1.00, 2.78) | 0.757 | 0.88 (0.38, 2.03) |
| No | 26 | 149 | 1 | | 1 | |
| **Know cervical cancer by visual inspection of the cervix** | | | | | | |
| Yes | 77 | 200 | 0.000 | 7.8 (3.28, 18.36) | 0.000 | **14.2 (3.77, 53.32)** |
| No | 6 | 121 | 1 | | 1 | |
| **Know premalignant cervical cancer by HPV DNA** | | | | | | |
| Yes | 39 | 110 | 0.033 | 1.7 (1.04, 2.77) | 0.397 | 0.69 (0.29, 1.61) |
| No | 44 | 211 | 1 | | 1 | |
| **Know the frequency of cervical cancer screen** | | | | | | |
| Yes | 66 | 196 | 0.000 | 2.8 (1.62, 4.99) | 0.690 | **0**.98 (0.42, 2.28) |
| No | 18 | 152 | 1 | | 1 | |
| **Know the WHO recommended cervical cancer screening age** | | | | | | |
| Yes | 55 | 179 | 0.021 | 1.8 (0.90, 2.94) | 0.941 | 1.0 (0.49, 2.16) |
| No | 29 | 169 | 1 | | 1 | |
| **Cryotherapy is treatment given for premalignant cervical lesion** | | | | | | |
| Yes | 66 | 204 | 0.001 | 3.6 (1.662, 5.78) | 0.195 | 1.8 (0.74, 4.34) |
| No | 13 | 123 | 1 | | 1 | |
| **Radiotherapy is treatment given for premalignant cervical lesion** | | | | | | |
| Yes | 43 | 125 | 0.009 | 1.9 (1.18, 3.17) | 0.043 | 2.1 (0.46, 2.16) |
| No | 36 | 202 | 1 | | 1 | |
| **Attitude of cervical cancer screen** | | | | | | |
| Favorable | 61 | 211 | 0.043 | 1.7 (1.02, 2.91) | 0.998 | 0.99 (0.46, 2.16) |
| Unfavorable | 23 | 137 | 1 | | | |

OPD=Outpatient department, OR= operation room, ART & TB = antiretroviral and tuberculosis treatment rooms

>15 years (AOR=5.4, 95% CI 1.09–26.36), respectively. Similarly, the study participants knowledge of the etiology (AOR=1.6, 95%CI=1.01–12.12), cigarette smoking as risk factor (AOR=4.1, 95%CI1.68–9.76), and VIA as screening method (AOR=14.2, 95%CI3.77–53.32) have shown an independent association with screening service utilization.

## Discussion

This study was conducted to assess the cervical cancer screening utilization and the factors affecting it among female healthcare workers in the public health institutions of AA, Ethiopia. Encouraging healthcare workers was reported as a significant predictor of the screening

uptake on top of implementing structured screening program in place [26]. The 19.4% prevalence of cervical cancer screening utilization among the total participants reported in our study is consistent with studies done in Nigeria, Iraq, India, and Kenya with the screening service utilization rates; 20.8%, 18.5%, 25%, 20%, and 25%, respectively [25,27–30]. The similarity could be explained partly by the fact that all the stated studies are from developing countries where cervical cancer screening is by and enlarge opportunistic and not well organized. The reported screening service utilization in the current study is higher compared to the previous reports from Ethiopia; The pooled national prevalence reported from two studies (8.1%, 14.8%), Southern Ethiopia (11.4%), Northern and Northwest Ethiopia (10.7%). This difference could possibly be explained by the difference in the study design and population; 1. The national pooled prevalence was obtained from systematic review and metanalysis of 44 and 25 studies, 2. The population from the Southern Ethiopia were not health professionals, and 3. The Northern Ethiopia study participants were only nurses, and those from the Northwest Ethiopia were recruited only from the hospitals [18,31–34]. As clearly depicted in the study participants section, our participants were recruited from the capital city of the country and comprised of the female healthcare providers from all health disciplines working at the tertiary and primary levels of the healthcare system of the country (referral hospitals and health centers).

The prevalence of screening utilization among the participants in the targeted age group (age ≥30 years) of 31.2% in our study is slightly higher compared to the previous study conducted in AA which was 25% [11]. This disparity could be possibly explained by the difference in the study population; the outcome in the previous study was a knowledge, attitude, and practice (KAP) and the study population were community health extension workers. Contrary to the above, the service utilization of 31.2% in our targeted study participants is lower than the report from Japan (54.7%), United States (71%), and Cameroon (43.48%). This variation could be due to the age difference in the study population; the participants age range for the Japanese, United states, and Camerron women was 20–30, 45–65, and 25–65 years, respectively. In addition, the availability of well-established screening system in place, the difference in the socioeconomic status and health literacy could explain the observed differences for participants being from high income countries like Japan and the United States. [32,35,36].

The odds of screening utilization in our study which was 2.3 times higher in women with good knowledge is comparable to the earlier studies from Ethiopia and Nigeria, where awareness and knowledge about cervical cancer was shown to have significant association with the screening service uptake [22,32,33].

The reported screening service utilization by nurses in our study, 22.3% and 64.3%, within the nursing profession and among the total screening service utilizers, was comparable to the report from the Nigerian study where the proportion of nurses was 22.7% [31]. The possible explanation for the larger proportion of service utilization by nurses in our study includes; 1. The nurses make the larger proportion among the healthcare workers who obtained the Federal Ministry of Health lead training on cervical cancer prevention, screening, and treatment, 2. They are the ones largely allocated to the screening unit, and essentially deliver the screening services, which gave them the opportunity for better awareness and service utilization.

The reported odds of the screening services utilization in our participants which was 3.7 times higher among women working in cervical cancer screening unit is in agreement with the 2020 systematic review and metanalysis report from Ethiopia, and the cross-sectional study done in Mekele town, Northern Ethiopia, where place of work was shown to have significant association with the screening service utilization [32,33,37].

The finding in our study that participants knowledge of the etiology of cervical cancer was associated with slightly higher screening service utilization with an odds of 1.6 is comparable to the study done in Mekele town, Northern Ethiopia [37]. Similarly, the odds of screening service utilization 4.1 and 14.2 for participants who knew cigarette smoking as risk factor and visual inspection with acetic acid (VIA) as screening method in our study is consistent with the systematic review and metanalysis report from Nigeria, where awareness about the screening methods was the positive facilitator of the screening utilization. Likewise, the odds of screening service uptake was reported to be 3.2 among participants with good knowledge about the screening methods in the previous studies from Ethiopia [22,32,33]. The independent association of the positive attitude with screening service uptake in the previous study from Ethiopia was only demonstrated on the crudes odds ratio (COR) in current study, and this could partly be explained by the difference in the design of the studies [33].

This study has specifically addressed the screening service utilization of all female healthcare workers in the public health facilities in the capital city of the country, where organized cervical cancer screening services being implemented. The participants being from the various health-related disciplines and who were working both at the primary and tertiary healthcare systems of the country were some of the strengths of this study. Since Addis Ababa is the capital city of the country and Ethiopia is a country with diverse ethnic and cultural differences, one can't simply generalize the results of the findings to all female healthcare workers practicing across all corners of the country. Too few observations (samples) in the regression model should be interpreted carefully as this might make it difficult to accurately estimate relationships between variables and potentially leading to misleading conclusions.

## Conclusion

The screening service uptake among total participants and the targeted age groups was far below the WHO cervical cancer elimination strategy for developing countries. The low screening service uptake was alarming since the study participants were female health workers who were expected to be role models to their community. Compounding the low screening service utilization by the health care workers was the lack of readiness to utilize the screening service even in the near future and there were some participants in our study who had never ever heard at all about cervical cancer screening.

Therefore, we urge the Addis Ababa Health Bureau and the Federal Ministry of Health of Ethiopia to consider this finding as an input in the future planning of cervical cancer prevention and control strategies. Particular attention needs to be paid to female healthcare workers in an effort to improve the cervical cancer screening uptake at the national level.

We further recommend a qualitative study to explore some masked behaviors of the female health care workers regarding the meagre screening uptake.

## Operational definitions

**Knowledge.** The understanding of the respondents regarding to carcinoma of the cervix with respects to symptoms, risk factors, screening methods, prevention and treatment.

**Attitude.** The belief and feeling of the respondents about screening for premalignant cervical lesions

**Practice/Utilized.** The action taken by individual respondents to go for screening/for herself weather screened or not. Those who had screened for herself considered as utilized.

**Knowledgeable.** We used 14 knowledge-based multiple-choice questions and participants who score above the mean value on knowledge questions were regarded as knowledgeable, and those who scored below the mean were regarded as not knowledgeable.

**Favorable Attitude.** We used 5 attitude-related questions and health professionals who score above the mean value on attitude questions were considered positive attitude towards cervical cancer screening.

**Barriers.** Circumstance or obstacle that keeps or hinders people action achieving, using or performing an activity.

**Uptake.** Alternatively used as Utilization in this context

**Utilization/utilized.** Those female health professionals who had screened for Premalignant cervical cancer.

## Supporting information

**S1 Fig. Multistage sampling procedure for cervical cancer screening utilization.**
(TIF)

**S2 Fig. Conceptual framework for factors influencing cervical cancer screening service utilization.**
(TIF)

**S3 Fig. A bar graph showing the main reason for study participants not screened.**
(TIF)

## Acknowledgments

We thank the Department of Obstetrics and Gynecology in the School of Medicine for allowing us to undertake this study. The female health workers who participated in this study deserve due recognition for their time and interest. Special thanks go to Girma Taye, (PHD, Head of Epidemiology and Biostatistics, CHS, AAU), and Mr. Mohamed Legas (MSc in reproductive health) for assisting us in statistical analysis. We are indebted to Lukman Yusuf, Professor Emeritus of Obstetrics and Gynecology for assisting us in editorial and grammatical work of the manuscript.

## Author contributions

**Conceptualization:** Shiferaw Negash Abebe, Achamyelew Melaku, Sofanit Haile.

**Data curation:** Shiferaw Negash Abebe, Achamyelew Melaku.

**Formal analysis:** Achamyelew Melaku.

**Funding acquisition:** Achamyelew Melaku.

**Investigation:** Achamyelew Melaku.

**Methodology:** Shiferaw Negash Abebe, Achamyelew Melaku, Sofanit Haile.

**Project administration:** Shiferaw Negash Abebe, Achamyelew Melaku.

**Resources:** Shiferaw Negash Abebe, Achamyelew Melaku.

**Software:** Achamyelew Melaku.

**Supervision:** Shiferaw Negash Abebe, Sofanit Haile.

**Writing – original draft:** Achamyelew Melaku, Sofanit Haile.

**Writing – review & editing:** Shiferaw Negash Abebe.

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
