## [Decision Letter · Decision Letter 0]

14 Jan 2025

PONE-D-24-37781Title: Utilization of screening services on cervical cancer and associated factors among female health workers in Addis Ababa, EthiopiaPLOS ONE

Dear Dr. Abebe,

Thank you for submitting your manuscript to PLOS ONE. After careful consideration, we feel that it has merit but does not fully meet PLOS ONE’s publication criteria as it currently stands. Therefore, we invite you to submit a revised version of the manuscript that addresses the points raised during the review process.

The review comments are detailed and self-explanatory. You may choose to accept all of these comments or disagree with a few of them, in either case, please be elaborate and rational to respond to them. 

We look forward to receiving your revised manuscript.

Kind regards,

Muhammad Farooq Umer, PhD Epidemiology and Health Statistics

Academic Editor

PLOS ONE

Journal Requirements:

“This manuscript was partially funded by the Addis Ababa University Office for  Graduate studies. The rest of the fund was equally shared among the authors.”

3. Please ensure that you refer to Figures 1 and 2 in your text as, if accepted, production will need this reference to link the reader to the figure.

4. We note that your Data Availability Statement is currently as follows: “All relevant data are within the manuscript and its Supporting Information files.”

Please confirm at this time whether or not your submission contains all raw data required to replicate the results of your study. Authors must share the “minimal data set” for their submission. PLOS defines the minimal data set to consist of the data required to replicate all study findings reported in the article, as well as related metadata and methods (https://journals.plos.org/plosone/s/data-availability#loc-minimal-data-set-definition ).

If your submission does not contain these data, please either upload them as Supporting Information files or deposit them to a stable, public repository and provide us with the relevant URLs, DOIs, or accession numbers. For a list of recommended repositories, please see https://journals.plos.org/plosone/s/recommended-repositories .

Reviewers' comments:

Reviewer's Responses to Questions

**Comments to the Author**

1. Is the manuscript technically sound, and do the data support the conclusions?

Reviewer #1: Partly

Reviewer #2: Yes

2. Has the statistical analysis been performed appropriately and rigorously? 

Reviewer #1: No

Reviewer #2: Yes

3. Have the authors made all data underlying the findings in their manuscript fully available?

Reviewer #1: No

Reviewer #2: Yes

4. Is the manuscript presented in an intelligible fashion and written in standard English?

Reviewer #1: Yes

Reviewer #2: Yes

5. Review Comments to the Author

Reviewer #1: Dear Editor, I would like to thank the team for invitation to review this manuscript and serve this prestigious journal. I think the manuscript has relevance to the local and national stakeholders. However, it needs major modification to fit the quality requirements for publication in this prestigious journal.

1.Line #1: which term is appropriate “utilization” or “use”

2.Line 22: what is the importance of depicting date and place here?

Abstract

Background

1.Line 31-33: “Healthcare workers, being the front line in health delivery system, are expected to play a critical 32 role in cervical cancer screening”. Do the authors believes that health workers should be role models in healthcare services utilization? Instead, it is better to state what makes health workers the major focus in this research? Are they at higher risk?

2.Line 33: do authors think that saying “Female” is necessary while talking about cervical cancer, no one can consider that males may take this service.

Method

1.Line 40-42: authors should differentiate the difference between “ Bivariate and Bi-variable”, also “Multivariate and Multi-variable”. Please use the correct terms elsewhere in the document.

Result:

1.50-52: better to show the figure.

2.Line 52-58: please put the AOR immediately after each respective variable.

3.Use appropriate spacing and typology.

4.Line 56: write the VIA in its full word.

Conclusion

1.What are the criteria to say low?

Introduction

1.Line 67: better to start with definition, and basic clinical and epidemiologic concepts of cervical cancer.

2.Line 99-100: clearly stated the uptake of CCS was low. knowing this, why the authors need additional study? This also contradicts with statement on line 32-33.

3.In Ethiopia which age group is a target for CC screening services? Better to introduce the CC screening program in Ethiopia.

Method

1.Line 135-137: write to the point, do not need elaborate extra information unrelated to the subsection. Only study design and period should be stated.

2.State eligibility criteria in detail under separate subsection

3.Line 125: why private health facility workers excluded? Authors should justify!

4.Line 133: add details related with cervical cancer screening services and female health workers in the study setting!

5.Prior to selection how the authors differentiates those eligible? For example how did you know their sexual activity? Cervical cancer status? Hysterectomy status? before selection and interviewing?

6.Are healthcare workers and health professionals the same?

7.Line 152: please state clearly how multi-stage sampling was applied, show each stage, stratifications, sampling units and selection processes.

8.Sample size calculation should consider adjustment on marginal error in line with the proportion of the outcome from previous study. Do you think using d=0.05 for p=15% fair? Usually, this margin of error corresponds with high proportion (40-60%)!

9.Line 160: why design effect of 2? how many stages do you have?

10.Line 161: write study variables under separate subsection entitled “Study Variables”

11.Line 164: show the allocation process clearly.

12.“Operational Definitions”, “Data collection Tools and Procedures”, “Data Quality and Management”…are missing. Please incorporate them!

13.Line 174-180: these statements are best suited under data collection subsection. Under data analysis section start your statement from data entry.

14.Line 184-186: authors should operationalize how they measured knowledge and attitude? Why mean was used?

RESULT

1.Line 199: how monthly income was categorized? Do you have reference?

2.Line 201: use the standard table and table titles. This comment is applied for all tables.

3.Is there a profession called “HO” ?

4.Why “HO” and “IESO” merged elsewhere?

5.Put specific numbers for “others” under the foot note.

6.Table 2 is not clear. if you need to show response for each knowledge question, write the question and show their responses.

7.Table 3 and 4 are poor quality and difficult to understand , should be modified to align the quality requirements of the journal.

8.Utilization of CC screening should be stated under separate section.

9.Indicate the interval estimate of the overall CC screening services utilization.

10.Line 163: the term Determinant” is not appropriate for this study.

11.Improve the way of reporting findings. Also do not use a long list followed by respectively, line 271-276.

12.Table 6 is poor quality, please improve it. Use consistent font style, color, bolding, size and spacing

13.The value (sample) in each cell should be adequate to fit the regression model. Unless it will cause numerical error. Please do any measurement to solve this problems. For e.g. marital status single and divorced contain too few values in the cells.

14.Interpret all statistically significant associations

15.You can also remove p-values, because CI is better explains the association.

Discussion

1.Line 297: use CI of the prevalence to compare your finding with others’.

2.Line 299: justify the possible explanations for the observed consistency in terms of the findings.

3.Line 303-310: how the difference in study design and population could result the observed discrepancy? How your study might differed from a study that included only nurse professionals?

4.Line 311-313: is not clear, “The prevalence of screening utilization among the participants in the targeted age group (age ≥30 years) of 31.2% is slightly high” stands for which study?

5.Line 313-315: the justification is not clear! How difference in primary objective caused the observed difference?

6.Line 317-321: do the authors think age difference is the only cause of the observed variation? I think many factors like sociodemographic, economic, health literacy, health setups..e.t.c. Used be considered.

7.Line 323-357: Interpretations should be stated under result. Under discussion section you need to compare and justify with other studies.

8.Line 327: explain how knowledge helps to increase utilization. Add this explanation for all associated factors with references.

9. Add the policy implication of your finding in Ethiopian context.

10.The authors should disclose the limitation of the study.

Conclusion

1.This section should be based on pertinent findings.

2.No need of numerical figures in this section.

3.Recommendations should consider based on major findings.

References

1.Check for completeness of all references. For example

a)ref#3, year of publication repeated.

b)Ref#16. is not consistent with Vancouver style.

Reviewer #2: Thanks for the chance to review this submission. The authors of this study investigated the cervical cancer screening utilization and associated factors among female health workers in Ethiopia. The study notion and findings are interesting and of importance. Please see below for my comments and suggestions.

1. Abstract: please edit according to journal requirements and world limits. Abstract could be much briefer.

2. Method, lines 161-164: these lines seem to be irrelevant to the sample size calculation. A dedicated section in the methods is needed to define all study variables and outcomes of interests.

3. Results, lines 274-275: “working in cervical cancer screening units showed an independent association with the screening service utilization”, this statement and finding would not be surprising as working in such centers is a confounder of the cervical cancer screening utilization.

4. Discussion: please add a paragraph on the study limitations.

6. PLOS authors have the option to publish the peer review history of their article (what does this mean? ). If published, this will include your full peer review and any attached files.

**Do you want your identity to be public for this peer review?** For information about this choice, including consent withdrawal, please see our Privacy Policy .

Reviewer #1: No

Reviewer #2: **Yes: ** Sina Azadnajafabad, MD, MPH

---

## [Author Response · Author response to Decision Letter 1]

18 Feb 2025

We the authors of this manuscript acknowledge the the comments given by the editor and the reviewers.

We are grateful for their time and constructive comments.

We don't have any specific comments other than the one addressed in the "response to editor's and reviewer comments"

---

## [Decision Letter · Decision Letter 1]

10 Mar 2025

Title: Utilization of screening services on cervical cancer and associated factors among female health workers in Addis Ababa, Ethiopia

PONE-D-24-37781R1

Dear Dr. Abebe,

We’re pleased to inform you that your manuscript has been judged scientifically suitable for publication and will be formally accepted for publication once it meets all outstanding technical requirements.

Kind regards,

Muhammad Farooq Umer, PhD Epidemiology and Health Statistics

Academic Editor

PLOS ONE

Additional Editor Comments (optional):

Reviewers' comments:

Reviewer's Responses to Questions

**Comments to the Author**

1. If the authors have adequately addressed your comments raised in a previous round of review and you feel that this manuscript is now acceptable for publication, you may indicate that here to bypass the “Comments to the Author” section, enter your conflict of interest statement in the “Confidential to Editor” section, and submit your "Accept" recommendation.

Reviewer #2: All comments have been addressed

2. Is the manuscript technically sound, and do the data support the conclusions?

Reviewer #2: Yes

3. Has the statistical analysis been performed appropriately and rigorously? 

Reviewer #2: Yes

4. Have the authors made all data underlying the findings in their manuscript fully available?

Reviewer #2: Yes

5. Is the manuscript presented in an intelligible fashion and written in standard English?

Reviewer #2: Yes

6. Review Comments to the Author

Reviewer #2: Thanks for the revision and amendments. I have no further comments or suggestions as the current manuscript is clear and scientifically robust.

7. PLOS authors have the option to publish the peer review history of their article (what does this mean? ). If published, this will include your full peer review and any attached files.

**Do you want your identity to be public for this peer review?** For information about this choice, including consent withdrawal, please see our Privacy Policy .

Reviewer #2: **Yes: ** Sina Azadnajafabad, MD, MPH

---

## [Editor Report · Acceptance letter]

PONE-D-24-37781R1

PLOS ONE

Dear Dr. Abebe,

I'm pleased to inform you that your manuscript has been deemed suitable for publication in PLOS ONE. Congratulations! Your manuscript is now being handed over to our production team.

Kind regards,

on behalf of

Dr. Muhammad Farooq Umer

Academic Editor

PLOS ONE